# Impact of Hypnotherapy on Fear, Pain, and the Birth Experience: A Systematic Review

**DOI:** 10.3390/healthcare12060616

**Published:** 2024-03-08

**Authors:** Laura Fernández-Gamero, Andrés Reinoso-Cobo, María del Carmen Ruiz-González, Jonathan Cortés-Martín, Inmaculada Muñóz Sánchez, Elena Mellado-García, Beatriz Piqueras-Sola

**Affiliations:** 1Virgen de las Nieves University Hospital, Av. de las Fuerzas Armadas, 2, 18014 Granada, Spain; laurafernandezgamero@gmail.com (L.F.-G.); maria.ruiz.gonzalez.sspa@juntadeandalucia.es (M.d.C.R.-G.); bpiquerassola@gmail.com (B.P.-S.); 2Department of Nursing and Podiatry, Faculty of Health Sciences, University of Malaga, 29071 Malaga, Spain; andreicob@uma.es; 3Research Group CTS-1068, Andalusia Research Plan, Junta de Andalucía, 18014 Granada, Spain; inmams1@correo.ugr.es (I.M.S.); emg2684@gmail.com (E.M.-G.); 4Department of Nursing, Faculty of Health Sciences, University of Granada, 18071 Granada, Spain; 5“La Chana” Health Center, Granada Health District, 18013 Granada, Spain; 6Costa del Sol Health District, Av. de Mijas, 28, 29640 Fuengirola, Spain

**Keywords:** hypnosis, hypnotism, hypnoanalysis, hypnotherapy, mesmerism, childbirth, delivery

## Abstract

In recent times, research has been conducted on the use of hypnosis during childbirth preparation and its effects on pain, fear, and overall childbirth experience. The main objective of this study was to analyze the published scientific literature on the use of hypnotherapy during childbirth preparation and the outcomes achieved during labor. A systematic literature review was conducted following the PRISMA 2020 protocol, with a search performed on the PubMed, Cinahl, Scopus, and WOS databases. Studies meeting inclusion criteria, including randomized controlled trials (RCTs), were evaluated for methodological quality using the PEDro scale. The searches yielded a total of 84 results, from which 7 RCTs of high scientific quality were selected. Each article examined the impact of a hypnosis intervention during pregnancy and the results obtained during labor. The analysis covered the use of epidural anesthesia, pharmacological analgesia during labor, self-reported pain, labor duration, type of delivery, fear of childbirth, and childbirth experience. The results demonstrated benefits in reducing fear and pain during labor, along with an enhancement in the overall childbirth experience. Hypnotherapy can be a valuable resource for reducing fear and pain during labor and improving the lived childbirth experience.

## 1. Introduction

Every woman desires a smooth childbirth experience and wishes to deliver a healthy baby. Childbirth should be a joyful moment and not a feared event for women. However, in today’s context, many women still excessively worry and feel anxious as they anticipate this moment [1]. The fear of childbirth has negative repercussions on women’s health and incurs additional costs in the healthcare system. These consequences underscore the importance of early detection and interventions to minimize the psychological and economic costs for pregnant women and the healthcare system [2].

Avoiding the negative impact of this fear on women’s health is a concern for all healthcare professionals. Badaoui, Kassm, and Naja (2019) estimated that 14% of women experience high levels of fear of childbirth (FOC), known as tocophobia. Elements of childbirth fear identified include fear of pain, fear of the unknown, fear of losing control, fear of having an episiotomy, fear for the baby’s life, and fear of the mother’s own ability to give birth [2]. Tocophobia ranges from mild to severe, and as its severity increases, it also affects the daily quality of life for pregnant women [2]. Studies on the frequency of childbirth fear have revealed that all women have some fear of childbirth, but severe fear is found in 20–26%, and 6–10% experience extremely severe fear (debilitating fear), impacting their quality of life [3]. Fear of childbirth is estimated to be higher in first-time pregnancy patients compared to those with previous pregnancies and childbirth history [3]. Although many reasons are attributed to fear of childbirth, the most common is reported as fear of pain during childbirth [3]. This fear can lead to elevated anxiety levels during pregnancy, associated with inadequate maternal adjustment to the motherhood role and a higher incidence of biological, mental, behavioral, and health issues in offspring [2]. Women suffering from tocophobia have psychologically vulnerable profiles and are more likely to experience childbirth as a traumatic event and develop postpartum post-traumatic stress disorder (PTSD) requiring professional help [2].

Furthermore, women with FOC are more likely to request cesareans or pain relief during labor [2]. The World Health Organization has stated that ideal cesarean rates should be between 10–15% since 1985, yet cesarean rates have increased worldwide in recent years, and fear of childbirth may be one of the causes. Studies have shown that fear of childbirth significantly affects birth outcomes and is one of the most important reasons for the dramatic increase in cesarean deliveries. The fear that the baby may suffer harm during normal childbirth is also a significant maternal concern, especially in developing countries. Therefore, pregnant women may, on some occasions, prefer cesarean sections, considering them safer for the baby and more comfortable due to avoiding labor pains [3].

Pain, especially during childbirth, is a complex phenomenon. At least four dimensions describe it: nociceptive (noxious stimuli), sensory–discriminative (intensity), affective–motivational (unpleasant emotional aspect), and cognitive–behavioral (behavior). Pain can be experienced differently from one person to another. This model helps professionals tailor their interventions to ensure that responses to pain are based not only on what they perceive in the behavior of the parturients but also on other dimensions affecting each woman’s experience with her pain [4]. According to the multidimensional conception of pain, a distinction can be made between pain and suffering.

Suffering in the context of childbirth is marked by a woman’s incapacity to engage her inherent pain relief mechanisms effectively or the insufficiency of these mechanisms to deal with the situation, leading to an unwelcome experience of pain. Effectively alleviating suffering requires healthcare professionals to comprehensively address the multiple dimensions of pain. Of particular significance is the affective–motivational dimension, given that emotional stress triggers the release of catecholamines, which could potentially prolong labor and escalate the need for obstetric interventions [4]. In this regard, managing emotional aspects becomes crucial for a holistic approach to pain relief during childbirth. By understanding and mitigating the affective–motivational dimension of pain, healthcare providers can contribute significantly to enhancing the overall childbirth experience and reducing the associated suffering [5].

The Society of Obstetricians and Gynaecologists of Canada “recommends non-pharmacological approaches as a safe first-line method for pain relief and should be continued throughout labor, whether pharmacological methods are used or not” [4]. After reviewing scientific evidence on non-pharmacological methods of pain management during childbirth, they issued a guideline that includes a series of recommendations whose quality was evaluated with the “Canadian Task Force on Preventive Health Care”. These recommendations consider the cognitive–emotional sphere of pain through supportive and non-pharmacological approaches for its management. Support measures should align with women’s desires, so working with them and listening to their needs is essential. Among its recommendations, the Society of Obstetricians and Gynaecologists of Canada stipulates that “to reduce the need for obstetric interventions and avoid associated risks and side effects, continuous support should be offered during labor with the addition of at least one other non-pharmacological pain-modulating mechanism and, as far as possible, promote and support the physiological progress of labor, childbirth, and puerperium by trusting in the woman’s ability to work with her pain and encouraging her to trust her ability to give birth” [4].

One of the recently studied methods that would support physiological progress of labor, increase women’s self-confidence in their ability to give birth, and alleviate fear and pain is hypnosis. Among many research studies, evidence regarding the efficacy of hypnosis in labor and childbirth is described in a recent Cochrane review (2016) of nine trials (*n* = 2954 randomly assigned women) testing the effectiveness of hypnosis for pain management during labor and childbirth [6,7]. Nevertheless, available scientific studies on the effectiveness of hypnotherapy for maternity care generally conclude that there may be benefits, but conflicting findings prevail [8].

According to the American Society of Clinical Hypnosis, hypnosis is defined as “a state of inner absorption, concentration, and focused attention during which a person responds largely to suggestion” [7]. Hypnotic communication is designed to project sensations and images into the patient’s consciousness that would induce relaxation and comfort. There is evidence that in hypnosis or hypnotic communication, the beneficial or negative effects of words can be enhanced [9]. Generally, the most commonly used types of hypnosis are Ericksonian hypnosis (71%), hypnotic relaxation therapy (55%), and traditional hypnosis (50%) [10]. Although most authors posit that hypnosis does not cause adverse reactions, some specify that in individuals with pre-existing psychotic mental pathology, there is a higher risk of exacerbating their underlying mental health problem [11].

The implications of using hypnosis in the medical field are extremely broad. Hypnosis has shown promising results in the treatment of depression, anxiety, anxious anticipation of medical interventions and pain, sleep disorders, obesity, nausea, vomiting, and self-efficacy, among other conditions [12]. Among the many variables that could influence hypnotherapy, most physicians rated the hypnotherapist–patient relationship (88%) and patient motivation (75%) as very important factors for success [10]. Healthcare professionals who effectively support women using self-hypnosis during childbirth must be trained and have skills in hypnosis, in addition to confidence in their own ability to facilitate this method, as previous research has established that self-efficacy is a strong indicator of performance [6].

The main theoretical model underlying the use of hypnosis for pain treatment during labor and childbirth is the fear–tension–pain syndrome model described by Dick-Read [13]. The model asserts that hypnosis can help women change their pre-existing beliefs about childbirth, resulting in increased confidence, lower anxiety, reduced muscle tension, and ultimately, reduced pain. Grantly Dick-Read, an English obstetrician and advocate for natural childbirth, believed that hypnotic relaxation could lower the level of panic and pain experienced by expectant mothers [14]. This model assumes that hypnosis has a direct effect on the sensory component of pain, i.e., the intensity of pain. Through different techniques, the pain threshold could be raised by a recalibration process, contributing to a lower perception of pain and, in turn, potentially impacting the use of pharmacological analgesics during childbirth [7].

However, certain difficulties arise in this research domain. Firstly, the mechanisms of action of hypnosis on pain remain unclear. Hypnosis could impact both the ability to deal with pain and the different components of pain. Secondly, interventions are heterogeneous (delivered by a therapist or self-induced), posing a problem when studies are grouped in meta-analyses. Moreover, self-hypnosis training courses are administered in different versions (organized in individual or group sessions, with a varied number of sessions, with or without a partner, sometimes with an audio recording for home practice). Lastly, the assessment of numerous relevant outcomes remains a challenge. In existing trials, primary outcomes relate to pain management, leaving the childbirth experience in a secondary position. However, these outcomes may be less relevant to patients [7].

In the field of obstetrics, the application of hypnosis in childbirth preparation is often popularly recognized as hypnobirthing training or the Mongan Method [15]. This training aids women in the prenatal period by preparing them for the upcoming birth, reframing the representation of childbirth from a painful and challenging experience to a non-threatening one. Simultaneously, it provides them with the opportunity to use this technique during childbirth to exert more control over pain and fear, giving them a sense of control over the situation and increasing their self-efficacy [12].

Hypnosis also fosters the bonding process, as it enhances awareness of their bodies and the bond with their babies, while empowering them with confidence in their ability to give birth [12]. Self-hypnosis is a crucial part of the intervention [12,16] as patients learn how to proceed with self-induced trance to prolong therapeutic gain, thereby enhancing a sense of independence and autonomy [12].

In conclusion, hypnotherapy improves the childbirth experience and postnatal well-being, provides better control of emotions during childbirth, and alleviates fear and pain associated with it. Including future parents in hypnosis protocols would also allow them to prepare for the upcoming birth and alleviate their potential fear of childbirth [12].

If we want to administer high-quality healthcare, it is necessary to consider a patient-centered care model. Considering the perspective, as well as respecting the values, preferences, and needs expressed by the patient, are key points. In this approach, ensuring physical and psychological comfort seems essential, emphasizing the importance of proper pain and discomfort management for patients to achieve comfort [7].

With this literature review, we aim to conduct a high-quality investigation that addresses the pressing need to identify effective strategies for enhancing obstetric healthcare and, consequently, women’s health during pregnancy, childbirth, and the postpartum period. The primary objective of this study is to contribute to this goal by deriving a set of premises that can serve to standardize concepts regarding the use of hypnotherapy during childbirth preparation and its impact on fear and pain during labor, the overall childbirth experience, and the clinical outcomes obtained therein. In terms of clinical outcome, the aim of this systematic review was to explore the effect of hypnotherapy on fear and pain at the time of birth.

## 2. Methodology

### 2.1. Review Protocol

The methodology used for the preparation of this report was a systematic review of the scientific literature published on the use of hypnotherapy during childbirth preparation and the outcomes during childbirth in terms of its impact on associated fear and pain, clinical results, and lived childbirth experience. The Preferred Reporting Items for Systematic Reviews and Meta-Analyses (PRISMA) review protocol was followed, which consists of a 27-point checklist covering the most representative sections of an original article, as well as the process of developing these guidelines.

### 2.2. Eligibility Criteria

We chose articles employing a randomized controlled trial (RCT) design, published from 2012 to the current date, that presented insights into the utilization of hypnosis as a therapeutic approach in childbirth preparation. The focus was on outcomes associated with fear and pain during childbirth. No constraints were imposed on the language of publication or the age of participants receiving the therapy.

### 2.3. Information Sources

The literature search was conducted in the Scopus, PubMed, Cinahl, and WOS databases. A manual search was also performed using reference lists of studies to find other relevant studies.

### 2.4. Structured Language

The structured language used was obtained through Medical Subject Headings (MeSH) terms and Health Sciences Descriptors (DeCS). The MeSH terms used were “hypnosis” and “parturition”, with the corresponding natural language terms “hypnotism”, “hypnoanalysis”, “hypnotherapy”, “hypnotherapies”, “mesmerism”, “parturitions”, “birth”, “births”, “childbirth”, and “childbirths”. Boolean operators used were OR and AND.

### 2.5. Search Strategy

The following table (Table 1) presents the search strategy used for this work, along with the date on which the search was conducted.

### 2.6. Data Extraction Process

After implementing the search strategy, the identified articles were transferred to the Mendeley web application using the Mendeley web importer tool. Subsequently, they were organized into folders based on the database from which they were obtained, and all duplicates were removed.

The included studies were randomized controlled trials (RCTs) aimed at evaluating the impact of hypnosis therapy used in childbirth preparation to improve data related to fear and pain during childbirth published between 2012 and 2024.

The author independently examined the title, abstract, and keywords of each identified study in the search and applied the inclusion and exclusion criteria. For potentially eligible studies, the same procedure was applied to full-text articles. Any doubts about a particular article were resolved through discussion with the project supervisor.

Data on quality, patient characteristics, interventions, and relevant outcomes were extracted by the author with the supervision of the project supervisor.

### 2.7. Data Collection Process and Collected Data

The author extracted the following data from each included article: author, year of publication, location of the study, participant characteristics (number, age, and sample characteristics related to the objectives), intervention characteristics (type, duration, frequency, and intensity of the intervention), study aim, and outcomes obtained. The strengths and weaknesses of each RCT were also assessed.

Section 3 provides a more detailed explanation of the article selection process.

### 2.8. Risk of Bias in Individual Studies

To conduct the methodological evaluation of the selected articles, an analysis of the design, methodology, and type of study for each work was performed to select the most specific methodological assessment scale for each case. Out of the 84 articles considered for full reading, 7 RCTs that addressed the researcher’s question were selected. Scientific quality assessment utilized the PEDro scale, a tool evaluating clinical scientific evidence.

### 2.9. Summary of Results

Based on the information provided by this review, a set of premises is obtained that will help standardize concepts regarding the use of hypnosis therapy during childbirth preparation and its impact on fear and pain during childbirth, the lived childbirth experience, and the clinical results obtained during the process.

## 3. Results

The searches yielded a total of 84 results, of which 7 RCTs were selected. The filtering and selection process can be observed in Figure 1.

The outcome of the methodological evaluation using the PEDro scale is shown in the following Table 2.

The scale assigns scores based on indicators, with one point added for each present indicator and zero points for absent indicators. Scores can range from 0 to 10, with 9 to 10 indicating very good quality, 6 to 8 denoting good quality, 4 to 5 signifying fair quality, and below 4 indicating poor quality. The selected articles for this systematic review received scores ranging from 7 to 9, yielding an average score of 7.85, indicative of a “good-quality” scientific standard. Table 2 displays the quality assessments for each RCT.

The RCTs included in this study were methodologically evaluated using the PEDro scale, obtaining an average rating of “good scientific quality.” A detailed analysis of the RCTs is observed in Table 3.

## 4. Discussion

With this systematic review, the aim was to obtain results that would help standardize concepts regarding the use of hypnosis during childbirth preparation and its impact on associated pain, fear, and the lived childbirth experience. In general, these results show that women who undergo hypnosis intervention during pregnancy experience a reduction in childbirth fear and an improvement in the childbirth experience. There is controversy regarding self-reported pain, duration and type of childbirth, and other clinically related data. No significant differences were found in the use of epidurals or other pharmacological analgesia.

Atis and Rathfish (2018) [17] conducted a study aiming to identify the effect of training with the hypnobirthing program during pregnancy on childbirth fear and pain. The results were very positive, highlighting that women who participated in the training experienced less pain and fear during childbirth. They affirmed that hypnobirthing reduces pain, and they showed a shorter duration of the second and third stages of labor, establishing early breastfeeding. Demographic tables indicated differences between the two groups, recognizing the difficulty in maintaining similarity in all sociodemographic variables.

Werner et al. published three different articles [18,19,21] studying different variables on the same sample. In a large RCT with 1222 participants, a brief prenatal course of hypnosis, inspired by Cyna, Bejenke, Waxman, and McCartny [18], contrary to the conclusions of Atis and Rathfish (2018) [17], had no effect on the duration of labor. It also had no effect on the frequency of vaginal delivery, the number of interventions, neonatal outcomes, or breastfeeding success [18]. In the same RCT, it was estimated that there were no differences between groups in the use of epidural analgesia, and no statistically significant differences were observed between the three groups for any self-reported pain measures [21]. Women in the hypnosis group experienced childbirth better than the other two groups, and this trend was also observed in subgroup analyses for type of delivery and fear levels [19]. In this regard, the results align with those of Atis and Rathfish (2018) [17]. Werner et al. (2013) [19] concluded that a brief course of self-hypnosis improved women’s childbirth experience.

The studies by Atis and Rathfish (2018) [17] and Werner et al. (2013) [19] suggest that hypnosis therapy may have a differentiated impact on the childbirth experience of nulliparous women (without previous childbirth experience) compared to those with previous experience. This differentiation could be attributed to various factors. In the case of nulliparous women, hypnosis might play a more significant role in addressing and reducing the fear of childbirth, providing effective tools for pain management, and fostering an overall more positive experience. Nulliparous women might be more receptive to hypnosis techniques because they do not have previous childbirth experiences influencing their expectations and perceptions.

On the other hand, women with previous experience may benefit from hypnosis differently, possibly using the techniques as a complement to their previous experiences. Hypnosis could help them enhance their perception of childbirth, adapt to new circumstances, or overcome potential previous traumas related to childbirth. However, previous experience can also influence how women with experience interpret and respond to hypnosis interventions, which could explain variations in outcomes.

Ultimately, the distinction in the impact of hypnosis between nulliparous and experienced women underscores the need for more detailed and specific exploration in future research, considering the diversity of experiences and needs of these two groups of women during the childbirth process.

Cyna et al. (2013) [20] conducted an RCT with 448 pregnant women in Australia to determine the use of pharmacological analgesia during childbirth when prenatal hypnosis is added to standard care. In this study, three different groups were compared: the intervention group with hypnosis + CD (compact disc), the CD-only group, and the control group. No differences were found in the use of pharmacological analgesia during labor and delivery in any group. There were no differences in key secondary outcomes regarding mode of delivery, use of labor stimulation, median duration of labor, or number of days of postpartum hospital stay. There were no significant differences between groups in admission to the high-dependency unit, the incidence of episiotomy, or the need for blood transfusion. There was also no difference in the incidence of exclusive breastfeeding at discharge, although more women in the control group reported exclusively breastfeeding at 6 weeks compared to those in the hypnosis group [20]. An increase in prostaglandin use for induction was noted in women assigned to the hypnosis group compared to controls. However, women exposed to hypnosis or CD stated that they would use hypnosis in future pregnancies. Subgroup analysis revealed that women who used yoga and received hypnosis used less analgesia than those who did not use yoga and received hypnosis [20].

Downe et al. (2015) [23], in their SHIP trial, observed that group self-hypnosis training did not significantly reduce the use of intrapartum epidural analgesia or a range of other clinical and psychological variables. However, according to the results obtained in the studies by Atis and Rathfish (2018) [17] and Werner et al. (2013) [19], women in the intervention group had actual fear and anxiety levels lower than expected from the onset to 2 weeks after childbirth, although postnatal response rates at 2 weeks were 67% overall. The additional cost in the intervention group per woman was GBP 4.83, and it was concluded that the impact of women’s anxiety and fear about childbirth needs further research [23]. The SHIP trial is the largest RCT on self-hypnosis for childbirth conducted in the UK to the date of completion. Except for a 2004 study conducted in the US, this trial is also the only one located in more than one center, including a range of births per site, and involves a large group of hypnosis practitioners, increasing the external generalization of findings. The study included birth partners, which does not seem to be the case in other trials in this area [23].

Finlayson et al. (2015) [22], in the UK, explored the views and experiences of a group of women receiving a prenatal hypnosis training program for childbirth pain relief (SHIP trial, Downe et al. (2015) [23]. Their results were encouraging, with most respondents reporting positive experiences of self-hypnosis and prominent feelings of calm, confidence, and empowerment. Participants found the intervention beneficial and used a variety of innovative strategies to customize their self-hypnosis practice. Occasionally, women reported feeling frustrated or disappointed when midwives misinterpreted their relaxed state upon admission or when their childbirth experiences did not match their birth expectations. Finlayson et al. (2015) [22] argued that the focused relaxation state experienced by women using the technique must be recognized by providers if the intervention is to be implemented in maternity services [22]. Among the described strengths, the authors pointed out that, besides a small study of six women conducted in Iran, this was the only published study exploring women’s views on using self-hypnosis during childbirth. Participants were randomly selected and representative of those in the index RCT, the procedures used for data collection and analysis were rigorous and transparent, and data interpretation was achieved by consensus among the research team [22].

In Malaysia, Beevi, Low, and Hassan (2017) [24] conducted an experimental study with a small number of women. They aimed to compare the results obtained in the experimental and control groups of primary variables measured during childbirth, such as the duration of the second and third stages of labor, analgesic use during labor, method of childbirth, and type of assisted vaginal delivery. Results of secondary variables, measured at 24 h postpartum, were also studied. These included newborn birth weight and the Apgar score at 1 min postpartum and at 5 min. Self-reported pain was also studied immediately before, during, and immediately after childbirth. In this study, more participants in the control group were administered epidurals, experienced assisted vaginal delivery, and underwent cesarean section. The mean neonatal birth weight was slightly higher, and a higher Apgar score was obtained in the experimental group. Despite experiencing higher levels of pain during labor, a smaller number of participants in the experimental group opted for pain relief. These results diverged from those obtained by Werner et al. (2013) [18,21], Downe et al. (2015) [23], and Cyna et al. (2013) [20] but agreed in stating that there were no differences in the duration of the second.

Although the systematic review suggests that hypnosis therapy may be helpful in improving the birth experience and reducing fear, there are discrepancies and limitations in the studies. Some authors disagree on the results on self-reported pain, duration, and type of labor. A large randomized controlled trial by Werner et al. found no significant effect of a brief course of prenatal hypnosis on the duration of labor or other clinical outcomes. Demographic differences between the study groups also complicate interpretation. Other studies present conflicting results on the use of pharmacological analgesia during labor with hypnosis interventions. The need for more quality clinical trials and standardization in interventions and in the endpoints and assessment tools used is highlighted. Research exploring women’s views on self-hypnosis adds a valuable perspective, highlighting the importance for health care providers to recognize the state of focused relaxation. In summary, although there are indications of benefits of hypnosis therapy in childbirth, more rigorous and standardized research is required to improve the reliability of findings in this field.

This article was registered at Prospero under the code CRD42024500572.

## 5. Conclusions

Hypnosis therapy can be a good resource to improve the birth experience and reduce fear of childbirth.

In the results obtained concerning the duration of the stages of labor, type of labor, and self-reported pain, there is controversy among the authors.

No differences between groups have been observed in the use of epidural analgesia and other clinical and psychological data related to childbirth.

More quality clinical trials, homogenization of interventions, variables to be studied, and tools used are needed to further advance this area of research.

## Figures and Tables

**Figure 1 healthcare-12-00616-f001:**
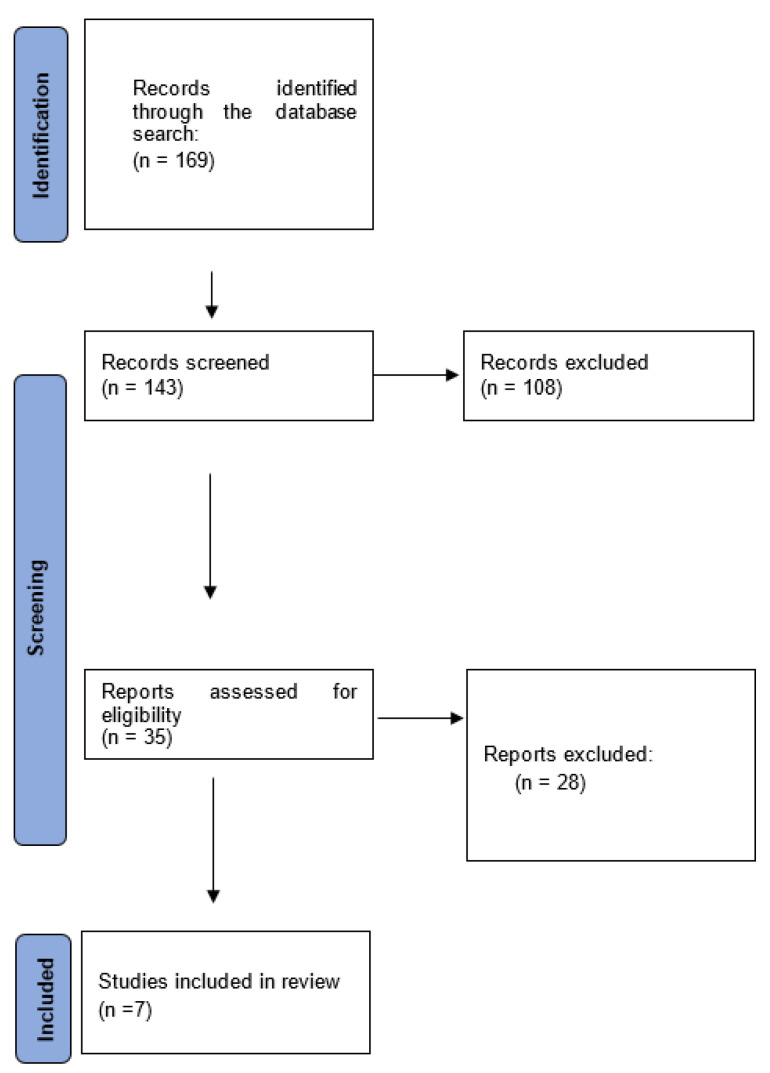
Flow diagram.

**Table 1 healthcare-12-00616-t001:** Search string.

Database	Search String
SCOPUS	((TITLE-ABS-KEY (hypnosis) OR TITLE-ABS-KEY (“hypnotism” OR “hypnoanalysis” OR “hypnotherapy” OR “hypnotherapies” OR “mesmerism”))) AND ((TITLE-ABS-KEY (parturition) OR TITLE-ABS-KEY (“parturitions” OR “birth” OR “births” OR “childbirth” OR “childbirths”)))
PUBMED	Search: ((hypnosis[MeSH Terms]) OR (“hypnotism”[Title/Abstract] OR “hypnoanalysis”[Title/Abstract] OR “hypnotherapy”[Title/Abstract] OR “hypnotherapies”[Title/Abstract] OR “mesmerism”[Title/Abstract]) AND ((y_10[Filter]) AND (clinicaltrial[Filter]))) AND ((parturition[MeSH Terms]) OR (“parturitions”[Title/Abstract] OR “birth”[Title/Abstract] OR “births”[Title/Abstract] OR “childbirth”[Title/Abstract] OR “childbirths”[Title/Abstract])
CINAHL	(MH “Hypnosis+”) AND (MH “Labor+”)
WOS	(TS=(hypnosis)) OR TS=(“hypnotism” OR “hypnoanalysis” OR “hypnotherapy” OR “hypnotherapies” OR “mesmerism”) AND (TS=(parturition)) OR TS=(“parturitions” OR “birth” OR “births” OR “childbirth” OR “childbirths”)

**Table 2 healthcare-12-00616-t002:** Results of the evaluation of RCTs with the PEDro scale.

ARTICLE	1	2	3	4	5	6	7	8	9	10	TOTAL	
Atis y Rathfish, 2018 [17]	YES	YES	YES	YES	¿?	¿?	YES	¿?	YES	YES	7	
Werner et al., 2013 [18]	YES	YES	YES	YES	YES	¿?	YES	¿?	YES	YES	8	
Werner et al., 2013 [19]	YES	YES	YES	YES	YES	¿?	YES	¿?	YES	YES	8	
Cyna et al., 2013 [20]	YES	YES	¿?	YES	YES	YES	YES	YES	YES	YES	9	
Werner et al., 2013 [21]	YES	YES	YES	YES	YES	¿?	YES	YES	YES	YES	9	
Finlayson et al., 2015 [22]	YES	YES	YES	NO	NO	SI	YES	¿?	YES	YES	7	
Downe et al., 2015 [23]	YES	YES	YES	NO	NO	SI	YES	¿?	YES	YES	7	
											AVERAGE	7857

**Table 3 healthcare-12-00616-t003:** Table of results according to PICOS declaration.

STUDY	PARTICIPANTS	INTERVENTION	COMPARISON	OUTCOME
Atis and Rathfish, 2018 [17]	Turkey. 60 participants. Experimental group (n = 30)|control group (n = 30)	Pregnant, primiparous, 20–36 weeks gestation, candidates for vaginal delivery, no prior illnesses, single fetus.	Experimental group: 3 h theoretical teaching weekly for 4 weeks. Control group: standard care. Second phase: experimental group—support in breathing, relaxation, imagination, and exercises. Control group: standard care.	Identify the effect of hypnobirthing training during pregnancy on fear and pain in childbirth. Women in the hypnobirthing group reported lower pain and fear during childbirth. They experienced calmness, relaxation, and better control. Shorter durations in the second and third stages of labor. Experimental group women initiated early breastfeeding. No differences in neonatal outcomes or Apgar scores.
Werner et al., 2013 [18]	Denmark. 1222 participants. Intervention group (n = 497)|active comparison group (n = 495)|control group (n = 230)	Nulliparous women, >18 years, Danish speakers, uncomplicated pregnancy, no chronic diseases, 27–30 weeks gestation.	Intervention group: 1 h self-hypnosis per week for three consecutive weeks. Active comparison group: Three 1 h prenatal classes on body awareness, relaxation, and mindfulness. Control group: standard prenatal care.	Examine the effect of a brief self-hypnosis course on labor duration and other outcomes. The self-hypnosis course had no effect on labor duration, mode of delivery, interventions, neonatal outcomes, or breastfeeding success. No reported adverse effects.
Werner et al., 2013 [21]	Denmark	Estimate epidural analgesia use and pain experienced during childbirth after a brief course in self-hypnosis.	No differences in epidural use. No statistically significant differences in self-reported pain measures. Fewer scheduled cesareans in the hypnosis group; more emergency cesareans. No significant differences in types of delivery.	There were no differences between the groups in the use of epidural analgesia. No statistically significant differences were observed among the three groups for any of the self-reported pain measures. The number of scheduled cesarean sections was lower in the hypnosis group, and the number of emergency cesarean sections was higher in this group. There were no significant differences between the types of delivery.
Werner et al., 2013 [19]	Denmark	Study the effect of hypnosis on the birth experience (secondary pre-specified outcome).	Women in the hypnosis group reported a better birth experience compared to the other groups. A brief self-hypnosis course improved women’s birth experiences.	Women in the hypnosis group experienced their childbirth better compared to the other two groups (average W-DEQ score of 42.9 in the hypnosis group, 47.2 in the relaxation group, and 47.5 in the usual care group (*p* = 0.01)). – The trend toward a better childbirth experience in the hypnosis group was also observed in subgroup analyses for the type of delivery and fear levels. – In this large randomized controlled trial, a brief course on self-hypnosis improved women’s childbirth experience.
Cyna et al., 2013 [20]	Australia. 448 women. Hypnosis + CD group (n = 154)| CD-only group (n = 143)|control group (n = 151)	Pregnant, 34–39 weeks gestation, candidates for vaginal delivery, cephalic presentation, single fetus.	Hypnosis + CD (guided by a hypnotherapist): three live prenatal hypnosis sessions plus corresponding audio CDs. CD only (administered by nurses): used the same CD as the hypnosis + CD group, but without hypnosis training. Control group: standard prenatal care, no additional CD.	Determine the use of pharmacological analgesia during childbirth when prenatal hypnosis is added to standard care. No differences in pharmacological analgesia use, epidural use, perceived pain, or satisfaction with the birth experience. No differences in secondary outcomes. Hypnosis group reported increased prostaglandin use for induction. No differences in high-dependency unit admission, episiotomy, blood transfusion, or post-birth hospital stay. Similar rates of exclusive breastfeeding at discharge, but more control group women exclusively breastfeeding at 6 weeks.
Downe et al., 2015 [23]	United Kingdom. 678 participants. Self-hypnosis group (n = 343)|control group (n = 335)	Nulliparous women, no planned elective cesarean, no hypertension medication, no mental illness, >18 years.	Self-hypnosis group: two 1.5 h training sessions between 32 and 35 weeks gestation, daily self-hypnosis CDs. Control group: standard prenatal care.	Establish the effect of prenatal group self-hypnosis on epidural use during childbirth. Secondary outcomes: clinical and psychological results, cost analysis. No statistically significant difference in epidural use or secondary outcomes. No difference in pain experience. Intervention group women had lower actual levels of fear and anxiety than anticipated.
Finlayson et al., 2015 [22]	United Kingdom. Self-hypnosis group (n = 343)|control group (n = 335)	Pregnant women, >18 years	Self-hypnosis group: two 1.5 h training sessions between 32 and 35 weeks gestation, daily self-hypnosis CDs. Control group: standard prenatal care.	Explore the views and experiences of women receiving a prenatal self-hypnosis training program for labor pain relief. Most respondents reported positive experiences with self-hypnosis, feeling calm, confident, and empowered. They found the intervention beneficial and used innovative strategies to personalize their self-hypnosis practice. Occasionally, frustration or disappointment was reported when midwives misinterpreted their relaxed state during admission or when labor experiences did not match expectations. The focused relaxation state experienced by women using the technique should be acknowledged by providers if the intervention is to be implemented in maternity services.

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
