# Peer review of "Impact of Hypnotherapy on Fear, Pain, and the Birth Experience: A Systematic Review"

_healthcare, 2024, doi:10.3390/healthcare12060616_

Round 1

Reviewer 1 Report

Comments and Suggestions for Authors

Dear authors,

Thank you for the opportunity to read the manuscript.

I read with interest the paper. It is well written and the topic is interesting.

I think that further studies are needed to have a major comprehension of the role of non-pharmacological support as hypnotic therapy and supportive psychotherapic counseling on both reduction of fear and improvement of childbirth experience and on pain management.

More studies are needed to evaluate the timing for the start of hypnosis during childbirth preparation, also in women with comorbidity or at higher risk of complicated or prolonged labor.

Conclusion section/conclusion sentence could be added. 

Limitation sections could be added.

Author Response

R1

Comments and Suggestions for Authors

Dear authors,

Thank you for the opportunity to read the manuscript.

I read with interest the paper. It is well written and the topic is interesting.

Thank you very much for your time and comments.

I think that further studies are needed to have a major comprehension of the role of non-pharmacological support as hypnotic therapy and supportive psychotherapic counseling on both reduction of fear and improvement of childbirth experience and on pain management.

More studies are needed to evaluate the timing for the start of hypnosis during childbirth preparation, also in women with comorbidity or at higher risk of complicated or prolonged labor.

 Indeed, I fully agree with you. That is what we are trying to do with our line of research.

Conclusion section/conclusion sentence could be added. 

Limitation sections could be added.

Conclusions section to be added

Limitations are added

Best regards

Reviewer 2 Report

Comments and Suggestions for Authors

LIne 77 and subsequent, please add a reference and better explain suffering, hat involves sensory and emotional area.

Please better specify the main outcome you want to study and the end of introduction. The main outcome will be for example pain reduction, anxiety reduction. 

Why the search was limited to ten years?

IN eleggibilità criteri upupa should better specify which outcome you decide to collect (i.e VAS, anxiety, and so on,  and with which scales, type of delivery, delivery duration , patient impression of change...)

Table 2 is a result, should be moved in that section.Moreover, please better descrive the items of PEDro scale.

LIne 245: that is the researcher question? Primary outcome? It should be defined at the beginning.

Please in result section you should better explain your result, in terms for example of fear reduction, pain control (has many studies and patients, with which hypnosis technique, and so on).From line 270 till the end, these I think are your results.

In discussion section you should discuss similarities and differences between the studies you included, your result and those obtained by previous revisions, the strength and also limits of your work, and at the end obtain the premise you search to standardise concepts of hypnosis. 

Comments on the Quality of English Language

English need minor revision.

Author Response

R2

Comments and Suggestions for Authors

Thank you very much for your comments

LIne 77 and subsequent, please add a reference and better explain suffering, hat involves sensory and emotional area.

Done

Please better specify the main outcome you want to study and the end of introduction. The main outcome will be for example pain reduction, anxiety reduction. 

Done

Why the search was limited to ten years?

The search is limited in time to provide a topical aspect to the study. In fact it has just been updated to the present date.

IN eleggibilità criteri upupa should better specify which outcome you decide to collect (i.e VAS, anxiety, and so on,  and with which scales, type of delivery, delivery duration , patient impression of change...)

The relevant changes are made

Table 2 is a result, should be moved in that section.Moreover, please better descrive the items of PEDro scale.

Check the PRISMA guide for systematic reviews and you will see that the methodological assessment of the selected articles should appear where it does. The items of the scale are better described.

LIne 245: that is the researcher question? Primary outcome? It should be defined at the beginning.

The line indicated belongs to the methodological assessment of the studies, it has nothing to do with what is requested, please check what you are asking for.

Please in result section you should better explain your result, in terms for example of fear reduction, pain control (has many studies and patients, with which hypnosis technique, and so on).From line 270 till the end, these I think are your results.

The results start on line 254. First you present the flow chart and then the results logging as indicated in PRISMA.

In discussion section you should discuss similarities and differences between the studies you included, your result and those obtained by previous revisions, the strength The requested changes are made and also limits of your work, and at the end obtain the premise you search to standardise concepts of hypnosis. 

The requested changes are made

Reviewer 3 Report

Comments and Suggestions for Authors

Dear authors,

Thank you for the submission. This review systematically reviewed the impact of hypnotherapy on fear, pain, and the birth experience among women.

However, there are still some issues that need to be further addressed.

1: Should adjust or remove background color in Table 1 to keep consistency.

2: This review studied publications between 2012 to 2022, it could be better to include the year 2023 to make the review more up to date.

3: In Table 2, the first line should be 1-10 instead of 2-11, please revise accordingly.

4: In the discussion part, authors should address the impact of hypnosis in nulliparous and experienced women as well.

5: It could be better to have a conclusion section after the discussion to summarize the main findings about hypnosis or hypnotherapy for women during childbirth.

Author Response

R3

Comments and Suggestions for Authors

Dear authors,

Thank you for the submission. This review systematically reviewed the impact of hypnotherapy on fear, pain, and the birth experience among women.

However, there are still some issues that need to be further addressed.

Thank you very much for your time and comments.

1: Should adjust or remove background color in Table 1 to keep consistency.

Done

2: This review studied publications between 2012 to 2022, it could be better to include the year 2023 to make the review more up to date.

The date of the review is updated to the present date, but the reality is that the articles in the review are not increasing as none of the publications in recent months meet the criteria for this work.

3: In Table 2, the first line should be 1-10 instead of 2-11, please revise accordingly.

Done

4: In the discussion part, authors should address the impact of hypnosis in nulliparous and experienced women as well.

Done

5: It could be better to have a conclusion section after the discussion to summarize the main findings about hypnosis or hypnotherapy for women during childbirth.

Done

Kind regards

Round 2

Reviewer 2 Report

Comments and Suggestions for Authors

Dear authors, the paper in my opinion was i much improved.

Anyway, I still believe that table 2 and risk of bias description of the included studies is a result, and therefore should be moved in that section (i.e. it is not a method, a section in which you describe how you search and classify studies) and only after you have retrieved them, you look for risk of bias.

Moreover, I think that a sentence on the main objective of this review in terms of clinical outcome should be specified at the end of the introduction, for example a systematic review was done to explore the effect of....since only after to explore the existing evidence you can start with  concept standardisation on hypnosis.

Finally, it would be better to follow the PICOS statement when you describe the RCTs included.(patient, intervention, outcome, comparators and so on).

Comments on the Quality of English Language

Just minor revision are needed.

Author Response

Comments and Suggestions for Authors

Dear authors, the paper in my opinion was i much improved.

Thank you very much

Anyway, I still believe that table 2 and risk of bias description of the included studies is a result, and therefore should be moved in that section (i.e. it is not a method, a section in which you describe how you search and classify studies) and only after you have retrieved them, you look for risk of bias.

Done

Moreover, I think that a sentence on the main objective of this review in terms of clinical outcome should be specified at the end of the introduction, for example a systematic review was done to explore the effect of....since only after to explore the existing evidence you can start with  concept standardisation on hypnosis.

Done

Finally, it would be better to follow the PICOS statement when you describe the RCTs included.(patient, intervention, outcome, comparators and so on).

Done

Best regards